# The Potential Use of *Nephelium lappaceum* Seed as Coagulant–Coagulant Aid in the Treatment of Semi-Aerobic Landfill Leachate

**DOI:** 10.3390/ijerph19010420

**Published:** 2021-12-31

**Authors:** Hamidi Abdul Aziz, Nur Syahirah Rahmat, Motasem Y. D. Alazaiza

**Affiliations:** 1School of Civil Engineering, Engineering Campus, Universiti Sains Malaysia, Nibong Tebal 14300, Malaysia; nursyahirahrahmat92@gmail.com; 2Solid Waste Management Cluster, Science and Technology Research Centre, Engineering Campus, Universiti Sains Malaysia, Nibong Tebal 14300, Malaysia; 3Department of Civil and Environmental Engineering, College of Engineering (COE), A’Sharqiyah University (ASU), Ibra 400, Oman; my.azaiza@gmail.com

**Keywords:** landfill leachate, coagulation–flocculation, *Nephelium lappaceum*, COD, solid waste

## Abstract

Chemical-based coagulants and flocculants are commonly used in the coagulation–flocculation process. However, the drawbacks of using these chemical materials have triggered researchers to find natural materials to substitute or reduce the number of chemical-based coagulants and flocculants. This study examines the potential application of *Nephelium lappaceum* seeds as a natural coagulant–coagulant aid with Tin (IV) chloride (SnCl_4_) in eliminating suspended solids (SS), colour, and chemical oxygen demand (COD) from landfill leachate. Results showed that the efficiency of *Nephelium lappaceum* was low when used as the main coagulant in the standard jar test. When SnCl_4_ was applied as a single coagulant, as much as 98.4% of SS, 96.8% of colour and 82.0% of COD was eliminated at an optimal dose of 10.5 g/L and pH 7. The higher removal efficiency of colour (88.8%) was obtained when 8.40 g/L of SnCl_4_ was applied with a support of 3 g/L of *Nephelium lappaceum*. When SnCl_4_ was utilised as a coagulant, and *Nephelium lappaceum* seed was used as a flocculant, the removal of pollutants generally improved. Overall, this research showed that *Nephelium lappaceum* seed is a viable natural alternative for treating landfill leachate as a coagulant aid.

## 1. Introduction

Landfilling is a very famous, recognised, and effective method for municipal solid waste management globally, owing to its low cost and simple operational mechanisms [1,2,3]. Landfill leachate is one of the major issues of landfilling method [3]. Leachate is usually generated by water infiltration through the waste; hence, it often includes suspended or dissolved solids from the disposed of materials. Generally, leachate is characterised in terms of pH, total suspended solids (TSS), dissolved oxygen (DO), chemical oxygen demand (COD), biological oxygen demand (BOD), total Kjeldahl nitrogen (TKN), ammonium nitrogen (NH_3_-N), heavy metals, and others [4]. The composition of the leachate and the standard effluent quality vary by landfill location and legislation. The composition of leachate is affected by a number of parameters, including waste content, local meteorological conditions, landfill physicochemical conditions, and landfill age [5]. The structure and characteristics of the leachate are the most important elements influencing treatment method selection [6,7].

Biological treatment has proved its efficiency in treating different leachate types [8]. However, for old leachate with high COD and ammonium concentrations, numerous heavy metals, and low biodegradability (high COD/ BOD ratio), the efficacy of this approach did not show good removal results [9]. On the other hand, old leachate was found to be removed better using chemical and physical methods as compared to young leachate [10,11]. In general, it is easy to predict the quantity of long-term leachate; however, since the precipitation affects the short-term leachate, it is difficult to predict its quantity [12].

Generally, at the early stages of leachate production from waste decomposition, leachate has a large content of BOD_5_, biodegradable and non-biodegradable compounds like volatile fatty acids [11]. However, the leachate from old landfills typically contains a significant percentage of non-biodegradable compounds such as fulvic and humic-like chemicals [13]. Furthermore, inorganic molecules such as NH_3_-N are produced by the hydrolysis and fermentation of nitrogen-containing fractions of biodegradable waste substrates in stabilised leachate formed from older landfills. [14]. When the waste is stabilised, and leachate is collected and transported for treatment, washout influences the accumulating concentration of NH_3_ [15].

Physicochemical methods are commonly employed to pre-treat or polish the leachate. Coagulation–flocculation is a well-known chemical technique for leachate pre-treatment [16]. This method is normally used to remove suspended particles and recalcitrant compounds like humic acids, heavy metals, polychlorinated biphenyls (PCBs), and absorbable organic halides from the leachate (AOX). The mechanism of the pre-treatment method primarily entails charge neutralisation between the negative charge colloids and cationic hydrolysis products, followed by impurity amalgamation by flocculation [17]. The main parameters removed by this method are total suspended solids (TSS) and colloidal particles [18].

Several materials are used as coagulants and flocculants in the coagulation–flocculation process, where inorganic metal salts are extensively used for this purpose. Excessive use of these inorganic metal salts, on the other hand, may have negative impacts on the environment and represent a health concern [19]. To lessen and remove the negative impacts on living beings and the environment, natural coagulants/flocculants are critical to be developed and applied for landfill leachate treatment.

Malaysia is a tropical country that has a wide variety of fruits. As a result, numerous food companies make canned fruits to take advantage of the availability. In Malaysia and Thailand, the *Nephelium lappaceum* canning industry is well-established, and it involves the manufacture of *Nephelium lappaceum* fruits in syrup [20]. The *Nephelium lappaceums* are deseeded during the canning process, and the seeds are generally discarded as waste by-products. It is, therefore, beneficial to value add the application of the *Nephelium lappaceum* seed from being just disposed of. Zurina et al. [20] used *Nephelium lappaceum* seed polysaccharide as a natural coagulant to remove water turbidity. However, to the author’s knowledge, no published research has specifically focused on the use of *Nephelium lappaceum* seed as a natural coagulant or flocculant in the treatment of landfill leachate.

The main objective of this study was to scrutinise the effectiveness of using *Nephelium lappaceum* seed as a natural coagulant in leachate treatment. Specifically, *Nephelium lappaceum* seed was used in two states; where the first state involved the *Nephelium lappaceum* seed as a sole coagulant, whereas the second state used the *Nephelium lappaceum* seed as a flocculant associated with Tin (IV) chloride (SnCl_4_). The application of *Nephelium lappaceum* seed combined with SnCl_4_ has not been used in leachate treatment to date. The target parameters were suspended solids (SS), colour, and COD, which generally present at high concentrations in many old landfill sites.

## 2. Materials and Methods

### 2.1. Sampling and Characterisation of Leachate

The case study leachate samples were taken from one of the old landfills in Malaysia, i.e., the Alor Pongsu Landfill Site (APLS). APLS is one of the existing landfills in Malaysia located in Bagan Serai, Perak, with coordinates of 5°05′00.6′′ N 100°35′53.1′′ E. APLS was constructed in 2000 as an open dumping site and categorised as a stabilised and matured landfill as it has been operating for more than 10 years [21]. The total area of APLS is about 10 acres where it consists of a large dumping site and three leachate collection ponds. These ponds are in an anaerobic condition which acts as raw collection ponds. Leachate samples were collected six times within four consecutive months, starting from January 2020 to April 2020. All procedures of sampling (grab method), storage, and preservation of landfill leachate followed the Standard Methods for the Examination of Water and Wastewater [22]. The pH, temperature, dissolved oxygen (DO), and total dissolved solids (TDS) were analysed on-site using YSI 556-Probe System. All samples were transported to the laboratory and maintained in a cool chamber at 4 °C for further use. This procedure is to minimise any reaction, including chemical and biological reactions that may affect leachate properties [2]. The COD, BOD_5_, NH_3_-N, colour, and SS were conducted in the laboratory immediately upon arrival.

### 2.2. Extraction and Characterisation of Nephelium lappaceum Seed

Fresh *Nephelium lappaceum* fruits were bought from a roadside store at Kulim, Kedah, Malaysia. The fruits were deseeded and washed using tap water before drying them in an oven at a temperature of 105 °C for 10 to 15 min. This is to ensure easy peeling off of the seed coat of *Nephelium lappaceum* seeds, thus preventing the increase of moisture content in the seeds that may damage its natural content. The seeds were then crushed into powder by using a domestic blender and kept in airtight containers until further use.

A 50 g package of *Nephelium lappaceum* seed powder was mixed with one litre of distilled water to produce 50 g/L of *Nephelium lappaceum* seed stock solution. After that, the mix solution was blended again for two minutes. The suspension was then filtered using muslin cloth in a beaker to become the stock solution before being characterised in terms of pH, zeta potential, molecular weight, particle size, surface morphology, and functional groups. The molecular weight of *Nephelium lappaceum* seed was determined by using Malvern Zetasizer Nano ZS. Surface morphology was conducted using ZEISS SUPRA 35VP Field Emission Scanning Electron. Meanwhile, determination of the functional group of *Nephelium lappaceum* seed was conducted using Perkin-Elmer System 2000 FTIR spectrometer. SnCl_4_ as the main coagulant was the analytical grade of pentahydrate, SnCl_4_.5H_2_O with 98% purity grade, supplied by Telaga Madu Sdn. Bhd. Malaysia.

### 2.3. Coagulation–Flocculation Experiment

A jar test apparatus was used to conduct the coagulation–flocculation procedures. The jar test device has six agitators together with 2.5 cm × 7.5 cm rectangular blades for stirring and mixing. The velocity of the instrument and mixing duration were controlled manually. Before the coagulation–flocculation process, samples of landfill leachate were taken from the storage chamber and placed outside until they were conditioned to room temperature. After that, the sample was agitated thoroughly to ensure uniform mixing. Six 1000 mL beakers were filled with 500 mL of agitated sample. The pH of the samples was adjusted with 3 M HCl or 3 M NaOH until the desired pH was obtained. The desired pH was determined according to the optimum dosage, as shown in the next section.

The coagulation–flocculation process involves three important stages: rapid mixing, slow mixing, and settlement. A combination of fast mixing (100 rpm in 8 min), slow mixing (30 rpm in 20 min), and a settlement for 30 min was used for the experiments using *Nephelium lappaceum* seed as the main coagulant [23]. For the studies employing SnCl_4_ as coagulant without and with *Nephelium lappaceum* seed as flocculant, the rapid mixing of 200 rpm in 1.5 min, slow mixing of 40 rpm in 20 min, and settlement for 20 min were used [24]. The removal efficiencies of key parameters such as SS, colour, and COD in landfill leachate samples were used to estimate the efficacy of the coagulation–flocculation process. Equation (1) [25] was used to compute the removal efficiencies of each pollutant: Removal (%) = [(C_i_ − C_f_)/C_i_] × 100(1)
where C_i_: is the initial concentration of the samples, C_f_: is the final concentration of samples.

### 2.4. Determination of Optimum pH and Coagulant Dosage

Preliminary tests were carried out to obtain the pre-determined dosage and pH conditions. A preliminary test for *Nephelium lappaceum* seed was carried out using various dosages (0.5–15 g/L) of *Nephelium lappaceum* on raw leachate, whereas for SnCl_4_, dosages between 3.5 to 17.5 g/L were used. Then, the pre-determined dosages obtained were applied in varied pH landfill leachate from 2 to 12 to determine the best pH value for both *Nephelium lappaceum* seed and SnCl_4_. Together with this experiment, a pH control experiment was also conducted by adjusting the pH of the sample without any coagulant. This control process was carried out to ensure that the removal of the pollutant was not made solely by acids or alkalis.

Once the optimum pH was determined, a new set of experiments were carried out to determine the optimum dosages of *Nephelium lappaceum* seed using different dosages varying from 0 (as a control) to 3 g/L. Meanwhile, different dosages of SnCl_4_ varying from 0 (as control) to 17.5 g/L have been applied to the samples with the optimum pH value of SnCl_4_.

A jar test experiment was conducted to evaluate the use of SnCl_4_ as a coagulant in conjunction with *Nephelium lappaceum* seed as a flocculant. These were undertaken at different SnCl_4_ and *Nephelium lappaceum* seed dosages. Different dosages of SnCl_4_ were used, ranging from 5.60 g/L to 8.75 g/L. For *Nephelium lappaceum* seed, the dosage was from 0 g/L to 4 g/L. The same operating conditions for a jar test as before were applied.

## 3. Results and Discussion

### 3.1. Landfill Leachate Compositions

Table 1 shows the main characteristics of the raw leachate of APLS. The leachate has a temperature in the range of 26.48 °C to 32.07 °C with an average of 29.47 °C, which is within the permissible limit of below 40 °C as stated by the Malaysia Environmental Quality Act 1974. The leachate has a pH of 8.04 to 8.90, with an average of pH 8.59. This represents an old type of leachate. A young landfill (up to 5 years) has a pH from 3.7 to 6.5, whereas matured landfill has a pH higher than 7. This is because young landfill leachate contains carboxylic acid and bicarbonate ions that contribute to the low pH [26]. The concentration of the dissolved oxygen was between 2.14 and 4.38 mg/L with an average of 3.17 mg/L. The amount of DO in landfill leachate is usually low due to the various chemical pollutants as well as the high temperature that may restrain the concentration of oxygen in the landfill leachate [27]. COD value was between 2532 and 4215 mg/L with an average of 3440 mg/L. This far exceeded the standard discharge limit of 400 mg/L [22]. COD level in APLS landfill leachate was slightly lower than that of the previous study [28]. This is due to numerous factors that may affect the COD level of the landfill leachate, like; landfill age, the solid waste compositions, site characteristics, and climate conditions [29]. Besides that, APLS is a matured and stabilised landfill as the leachate contains a higher COD level, which is below 4000 mg/L [21]. BOD_5_ of landfill leachate from APLS ranged from 160 to 333 mg/L with an average of 241 mg/L., also exceeding the standard discharge limit of 20 mg/L. It can be concluded that the APLS is a stabilised landfill site as the BOD_5_ is less than 4000 mg/L [30]. The BOD_5_/COD ratio in leachate was about 0.070 within the range of 0.057 to 0.080. In general, a mature and stabilised landfill has a lower fraction of BOD_5_/COD, which is less than 0.1 due to the lower biodegradable fraction of organic pollutants in mature landfill leachate [21]. The low BOD_5_/COD ratio indicates that it would be difficult to treat the leachate using a biological process. NH_3_-N in APLS leachate was in the range of 1040 to 1357 mg/L and is slightly lower than the previous work [31]. The SS ranged from 350 to 604 mg/L and fell within the range of mature leachate. The leachate is negatively charged with a zeta potential reading of −21.5 mV; hence the majority of particles in leachate have the tendency to repel each other. Therefore, a coagulant with zeta potential higher than +21 mV is needed to neutralise the charges of particles in leachate and aggregate them into large flocs [12]. A recent study by Ramli et al. [32] showed similar results for the leachate characteristics. Figure 1 shows the particle size distribution of the leachate sample.

### 3.2. Characteristic of Nephelium lappaceum Seeds

*Nephelium lappaceum* seed has an acidic condition (ranging between 3.56 and 5.36) with an average pH of 4.45 ± 0.71. Generally, *Nephelium lappaceum* is an acidic fruit. According to Arenas et al. [33], *Nephelium lappaceum* has a pH range from 4.6 to 5.5, of which the pH was similar to its seed. The Zeta potential of *Nephelium lappaceum* seed was recorded to be about 5.46 ± 0.615 mV at its natural pH. By knowing its zeta potential, the *Nephelium lappaceum* seed is a cationic coagulant which means that it has a positively charged particle surface. Figure 2 shows the particle size distribution of *Nephelium lappaceum*.

The molecular weight of *Nephelium lappaceum* seed was measured at 3850 kDa. It was slightly higher if compared to Longan seed, where its molecular weight was 3280 kDa [16]. The increase in molecular weight of the coagulant, according to Ebeling et al. [34], encourages more particles to adhere to each other by providing extended branches and channels, resulting in larger flocs in the coagulation–flocculation process. Nevertheless, different coagulants may have their own coagulation–flocculation behaviour despite their high molecular weight [6].

The functional group of *Nephelium lappaceum* seed was determined using an FTIR Spectrometer, and the results are given in Figure 3 and Table 2. It is shown that *Nephelium lappaceum* seed contains many functional groups such as alcohol (O-H), amide (N-H), and carboxylic groups (C=O). However, only certain functional groups play an important role in coagulation–flocculation processes. According to Zafar et al. [35], natural coagulant neutralises the particle charges and aggregates flocs through bridging mechanisms by bio-polymers such as polysaccharides which can be found naturally in fruits and vegetables. Polysaccharides are one of the carbohydrate subgroups that contain carbon, hydrogen, and oxygen [36]. Furthermore, Aziz and Sobri [37] reported that carboxyl (C=O), hydroxyl (O-H), and amino (amine or amide) grouped together with polymeric chains have the capability to enhance flocculation through bridging. From the results, the content of (O-H), amide (N-H), and carboxylic groups (C=O) in *Nephelium lappaceum* seed are expected to serve as ion bridge or binding sites in coagulation–flocculation processes.

The surface morphology of *Nephelium lappaceum* seed is shown in Figure 4. According to FESEM micrographs, it can be seen that the *Nephelium lappaceum* seed surface has non-porous traits and mostly appears to be in globular shape. The surface morphology of *Nephelium lappaceum* seed appears similar to the Cassava peel, whereby it is smooth and globular in shape that covers the surface of Cassava peel [38]. It appears that most of the surfaces that contain starch are globular in shape due to the starch granules. Polysaccharides in the *Nephelium lappaceum* seed usually can be hydrolysed to hexose component, which is also known as starch. Moreover, *Nephelium lappaceum* seed has traces of alkaloids, sugar (1.25%), starch (25%), and ash (2%). Starch is considered as one of the biopolymers that are needed in coagulation–flocculation as it comprises a mixture of two polymers of anhydrous glucose units and amylopectin. These biopolymers can enhance the mechanism of adsorption, charge neutralisation, as well as perform inter-particle bridging in coagulation–flocculation.

### 3.3. Zeta Potential and Particle Size

According to Omar et al. [39], pH affects the zeta potential and aggregation of particles. For this, the zeta potential and particle size of leachate and *Nephelium lappaceum* seed solution were studied throughout a pH range of 2 to 12. Figure 5 shows the zeta potential of leachate, which was decreased from −9.53 mV to −29.5 mV as pH increased from pH 2 to 12, respectively. This trend indicates that landfill leachate was completely affected by pH. This observation is possibly due to the swift ionisation of both OH^−^ and H^+^ on the surface of particle size when the leachate pH was adjusted using acid (HCl) and alkali (NaOH) [39]. At pH 8, the zeta potential was about −22.4 ± 1.95 mV, and this signified that the particle surface of leachate was naturally negatively charged. Thus, to neutralise the leachate surface charge, a positively charged coagulant with almost the same value as leachate particles was needed, and thus, the coagulation–flocculation process will then be highly effective [40,41].

Figure 6 shows the relationship between the particle size of leachate and pH. Similar to zeta potential, leachate particle size also dropped as the pH increased. However, when the landfill leachate was added with 0.5 M NaOH in order to increase the pH from pH 8 to pH 12, the particle size subsequently decreased from 228.1 to 207.1 d.nm. According to Cao, et al. [39], the reaction of negatively charged ion, OH^−^ from NaOH, has interfered with the destabilisation of particles in the leachate by repelling each other and thus, made it impossible to form larger flocs. The particle size of landfill leachate increased from 228.1 d.nm to 9520 d.nm when the pH of the leachate was altered to lower pHs, which varied from pH 8 to pH 2. This is most likely due to the fact that when the solution was acidic, the density of H^+^ hydrolysates was extremely high, thus, dominate the charge neutralisation of the particles and forming a larger floc [39].

Kruszewski and Cyrankiewicz [42] stated that the aggregating agents (KCl or HCl) caused a substantial increase in particle size intensity as chloride was presented in these aggregating agents, which is capable of promoting the aggregation process and thus enhancing the flocculation process. Hence, when the pH of leachate increased, the particle size also increased. Characterisation of *Nephelium lappaceum* seed in terms of zeta potential and particle size as a function of pH were performed to evaluate the suitability of *Nephelium lappaceum* seed to be used as a natural coagulant. The association between pH and zeta potential, as well as particle size of *Nephelium lappaceum* seed, are depicted in Figure 1 and Figure 2, respectively. It can be seen that the trend of zeta potential of *Nephelium lappaceum* seed was also decreased as pH increased, which was identical to the behaviour of the leachate zeta potential trend. Between pH 2 to pH 6, the zeta potential of *Nephelium lappaceum* seed was observed to be in the positively charged surface, which was between 10.47 to 0.20 mV, respectively, and it was near to the point of zero charges (PZC). With a further increase of pH, which was from pH 7 to pH 12, the zeta potential decreased drastically from −8.00 to −20.1 mV. This signifies that *Nephelium lappaceum* seed became negatively charged surface after pH 6. During the coagulation–flocculation process, neutralisation of negatively charged surface of landfill leachate required positively charged coagulant.

The trend of *Nephelium lappaceum* seed particle size with respect to pH was opposite to landfill leachate particles size. Overall, the particle size of Nephelim lappaceum seed increased as the pH increased. However, at pH 2 to 5, the *Nephelium lappaceum* seed particle size decreased from 260.0 to 162.7 d.nm. This might be due to the fact that some negatively charged surface in the solution attracted with the positively charged ions that were hydrolysed from HCl acid [42], which was added in the *Nephelium lappaceum* solution to decrease the pH and thus, this had promoted the flocculation of particles. Furthermore, when pH increased from its natural pH until pH 12, the particle size was observed to increase from 162.7 to 2690 d.nm. Supposedly, the particles would scatter when NaOH was added into the sample due to the negatively charged ions of OH-. However, in *Nephelium lappaceum* seed with a natural positively charged surface, it would promote aggregation as NaOH has strong OH-ions that would help to form the inter-bridge between particles [43]. Based on the results of zeta potential and particle size of *Nephelium lappaceum* seed solution, it can be concluded that it is wise to choose *Nephelium lappaceum* seed at pH less than 6 because when the pH is high (pH 7–12), the *Nephelium lappaceum* seed has negatively charged surface and large particle size. This will affect the coagulation–flocculation process as the negatively charged surface of *Nephelium lappaceum* seed will repel the negatively charged surface of landfill leachate.

### 3.4. Optimum Operating Conditions of Nephelium lappaceum Seed as a Sole Coagulant

#### 3.4.1. Optimum pH

The effect of pH of *Nephelium lappaceum* seed as a natural coagulant in the coagulation–flocculation process was evaluated. A preliminary test was carried out first on raw leachate by using different dosages (0.5–15 g/L) of *Nephelium lappaceum* seed to find the pre-determined dosage used in determining the optimum pH of *Nephelium lappaceum* seed. Figure 7 demonstrates the effect of varying pH on the coagulation–flocculation process using a pre-determined dose of *Nephelium lappaceum* seed (1.5 g/L) and a pre-determined dosage of *Nephelium lappaceum* seed (1.5 g/L). At pH 2, the removal efficiency of SS, colour, and COD was 96.4%, 90.3%, and 56.4%, respectively. According to Bruice [43], acids are substances that can transfer a proton to a base and are known as proton donors. Therefore, at lower pH, a high concentration of H^+^ due to the hydrolysation of HCl has neutralised negatively charged particles in the landfill leachate. The charge neutralisation and complex reaction in the process were dominated by hydrogen ions from HCl, and hence, the suspension, together with pollutants, began to flocculate [44]. Beyond pH 4, the removals of all parameters were fairly low (less than 30%). This could be explained by the fact that *Nephelium lappaceum* seed naturally has a milky colour. Beyond pH 3, *Nephelium lappaceum* seeds also lose the effectiveness to remove SS. This observation could be explained by the fact that the *Nephelium lappaceum* seed solution has a large particle size and higher SS in the solution, and thus, this has added the quantity of SS in leachate. In conclusion, the *Nephelium lappaceum* was ineffective when used as the sole coagulant. Nonetheless, pH 6 was observed to be better for all the parameters. In conclusion, the best and optimum pH value for *Nephelium lappaceum* seed to remove SS, colour, and COD was at pH 6 whereby it removed 19.2% COD and 21.8% colour. Table 3 shows a comparison of previous studies that used natural coagulants in landfill leachate treatment with the current studies.

#### 3.4.2. Optimum Dosage

After the optimum pH value was obtained, the optimum dosage of *Nephelium lappaceum* seed was determined. In this study, various dosages of *Nephelium lappaceum* seed within the range of 0 to 3 g/L were employed to attain the optimum dosage at optimum pH value, which was pH 6 that was determined earlier. Figure 8 depicts the removal effectiveness for all the parameters. The results demonstrated that COD was the best pollutant removed (35% reduction at 2 g/L) by *Nephelium lappaceum* seed compared to others. The overall performance of *Nephelium lappaceum* seed alone as a coagulant was considered to be fairly low for all the pollutants. In summary, COD and colour removal were at their maximum performance at dosage 2 g/L, but *Nephelium lappaceum* seed could not remove SS from landfill leachate at this dosage, although it could remove 5.6% of SS at dosage 1 g/L. Therefore, the best and optimum dosage of *Nephelium lappaceum* seed to remove the pollutants from landfill leachate was at dosage 2 g/L. Table 4 displays the comparison of *Nephelium lappaceum* seed with other natural coagulants from previous studies.

From the comparison above, it can be seen that *Nephelium lappaceum* seed has the same dosage as Longan seed in removing pollutants, i.e., SS, colour, and COD effectively with medium dosage, whereas other coagulants such as commercial sago starch, durian seed starch, and *Tamarindus indica* seed needs slightly higher dosages. Nevertheless, among all-natural coagulants, *Nephelium lappaceum* seed demonstrates the best COD removal in landfill leachate. This is probably due to its positive zeta potential, as well as the presence of biopolymer in the seed that may enhance the inter-particles bridging between the seed and landfill leachate that ensures excellent removal of COD excellently.

### 3.5. Optimum Operating Conditions of SnCl_4_ a Sole Coagulant

#### 3.5.1. Optimum pH

The effectiveness of SnCl_4_ at different pH was investigated with the same procedures as before. Various dosages of SnCl_4_ (3.5–17.5 g/L) were examined. The pre-determined dosage of SnCl_4_, which was 10.5 g/L was employed at a broad range of landfill leachate pH from 2–12. Based on the results, the highest removal of SS, colour, and COD was noted at pH 7, whereby it removed 98.4% of SS, 96.8% of colour, and 8.0% of COD. According to Coffman [47], inorganic metal salts were normally found between pH 5 to 7. Yong and Aziz [46] found out that the optimum pH of PAC was pH 6, and Coffman [47] found that the optimum pH for titanium dioxide (TiO_2_) was pH 5 for landfill leachate treatment.

#### 3.5.2. Optimum Dosage

After obtaining the optimum pH value for SnCl_4_, the dosage of SnCl_4_ was then determined by using various dosages of SnCl_4_ within the range of 0 to 17.50 g/L, and the results are presented in Figure 9. The removals for all the pollutants sharply increased from low dosages until they reached a steady state with 98%, 85% and 83% reductions of SS, colour and COD, respectively, at 10.5 g/L SnCl_4_. Beyond this point, the removals dropped slightly. This is due to charge reversal, re-dispersal, and re-stabilisation of colloidal particles [43,44,45] and the overdosage phenomenon where the SnCl_4_ would cause the particle to re-disperse.

### 3.6. Performance of SnCl_4_ as a Coagulant and Nephelium lappaceum Seeds as a Flocculant

SnCl_4_ was used as a coagulant in this study, with the help of *Nephelium lappaceum* as a flocculant/coagulant aid. At the same time, the potential of *Nephelium lappaceum* seed assisting in reducing the coagulant dosages was also tested at a reducing rate from 8.75 g/L to 1.0 g/L of SnCl_4_. The efficiency of SnCl_4_ as a coagulant and *Nephelium lappaceum* seed as a flocculant was compared to the optimum and best dosage of SnCl_4_ as a sole coagulant, which was 10.50 g/L.

Figure 10a illustrates the colour removal efficiency. At 8.40 g/L of SnCl_4_ and 3 g/L of *Nephelium lappaceum* the removal for colour was about 89%. This is just slightly lower than 92% reduction when SnCl_4_ was used alone at higher concentrations (10.5 g/L). Not much significant difference was noted for SS (Figure 10b), where nearly 90% reduction was observed compared with 96% when SnCl_4_ was used alone. A slight reduction was due to the fact that *Nephelium lappaceum* seed particles are large in size [48,49], which renders them less effective in removing SS from landfill leachate [50]. Moreover, as discussed earlier, *Nephelium lappaceum* seed becomes negatively charged after pH 6. This might affect the particles charge neutralisation. Therefore, the removal of SS from landfill leachate mostly depended on SnCl_4_. Further lowering of SnCl_4_ dosages would only cause deterioration in treatment performance [51].

However, for the case of COD (Figure 10c), the performance of *Nephelium lappaceum* as coagulant aid was not so significant. The removal with and without *Nephelium lappaceum* was 76% and 89%, respectively. Generally, introducing *Nephelium lappaceum* seed as a flocculant with increasing dosage resulted in further reduction of COD removal efficiency [52]. This is because *Nephelium lappaceum* is an organic matter that contributes to the COD readings.

The summary of the findings is plotted in Figure 11.

## 4. Conclusions

In this study, *Nephelium lappaceum* seeds were used as an alternative coagulant for leachate treatment and compared with *Nephelium lappaceum* seeds as a flocculant associated with Tin (IV) chloride (SnCl_4_) as a main coagulant to remove suspended solids (SS), colour, and COD from landfill leachate. Results showed that *Nephelium lappaceum* was not effective when it was used as a sole coagulant. The best-operating conditions in terms of pH and dosage were found at pH 6 and 2 g/L, respectively. The removal of colour and COD were 19.5% and 35.7%, respectively. At pH 7, 10.5 g/L of SnCl_4_ exhibited a good performance as a sole coagulant with 98%, 85%, and 63% reductions respectively for SS, colour and COD. At 8.40 g/L of SnCl_4_ and 3 g/L of *Nephelium lappaceum* as flocculant, the removal for colour was about 89%, just slightly lower than 92% reduction when SnCl_4_ was used alone. No significant difference was noted for SS where nearly 90% reduction was observed compared with 96% when SnCl_4_ was used alone. However, *Nephelium lappaceum* as coagulant aid was not so significant for COD. With further investigations and optimisation work, *Nephelium lappaceum* could be a potential option to be an alternative to metal salt coagulant in treating landfill leachate.

## Figures and Tables

**Figure 1 ijerph-19-00420-f001:**
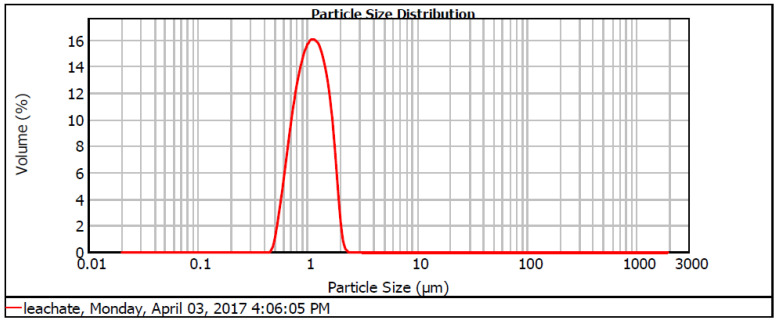
Particle size distribution for (a) landfill leachate.

**Figure 2 ijerph-19-00420-f002:**
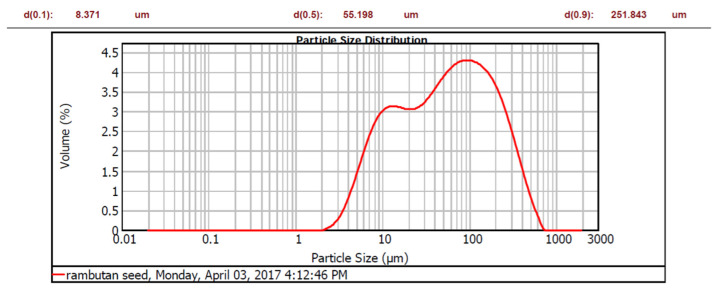
Particle size of *Nephelium lappaceum* seeds.

**Figure 3 ijerph-19-00420-f003:**
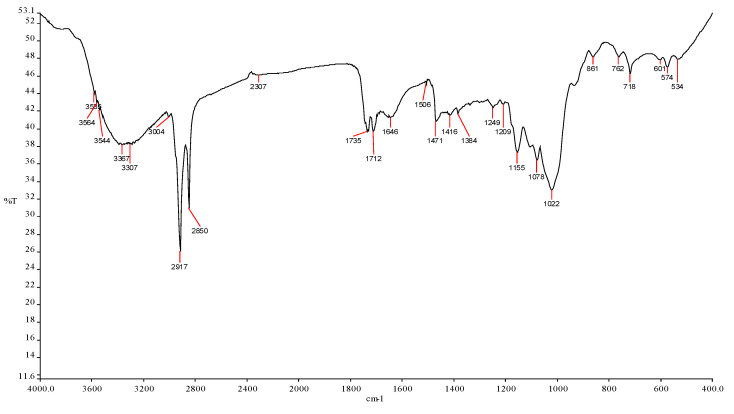
FTIR spectra of *Nephelium lappaceum* seed.

**Figure 4 ijerph-19-00420-f004:**
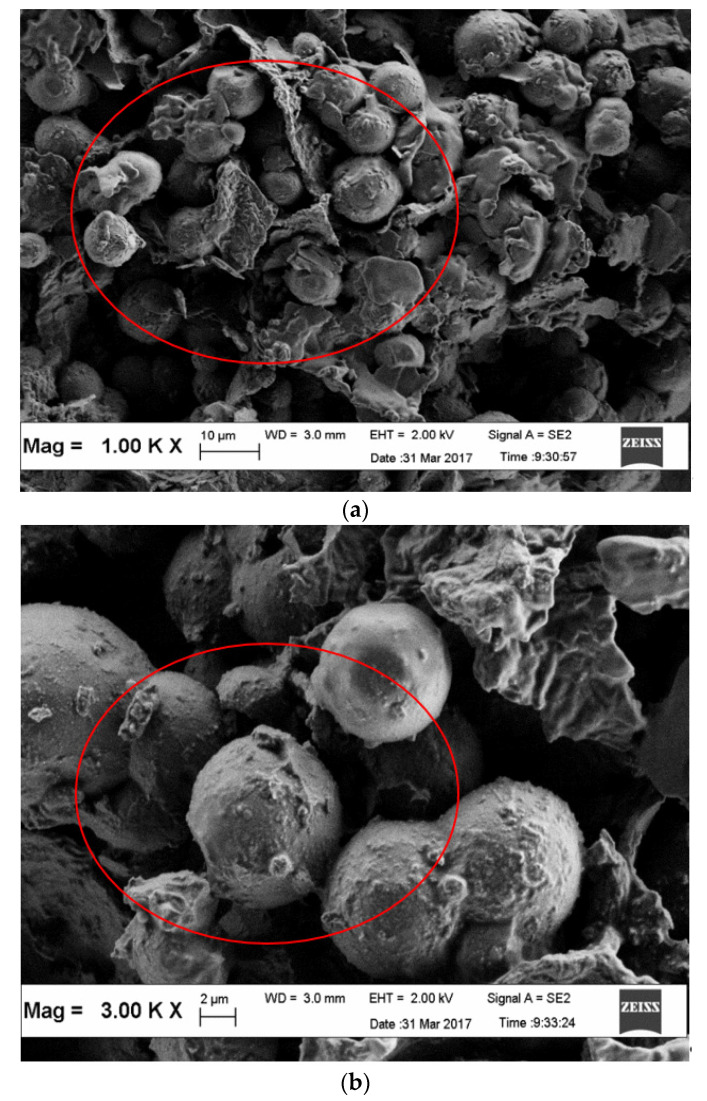
FESEM of *Nephelium lappaceum* seed. (**a**) 1000× magnification; (**b**) 3000× magnification.

**Figure 5 ijerph-19-00420-f005:**
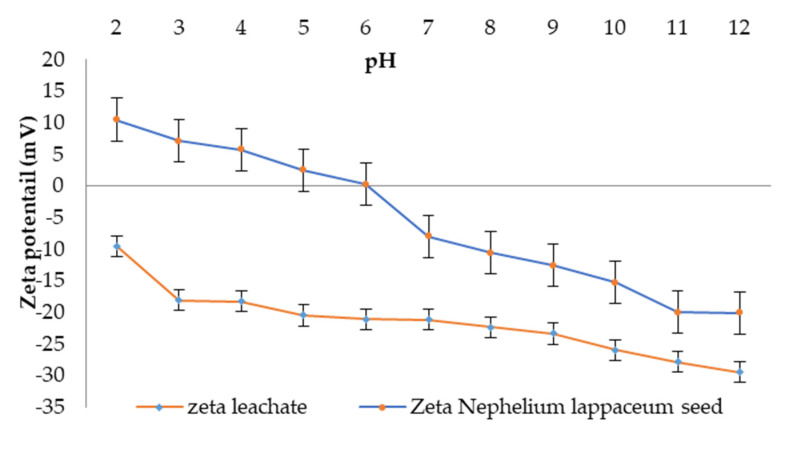
pH effect on Zeta potential of leachate.

**Figure 6 ijerph-19-00420-f006:**
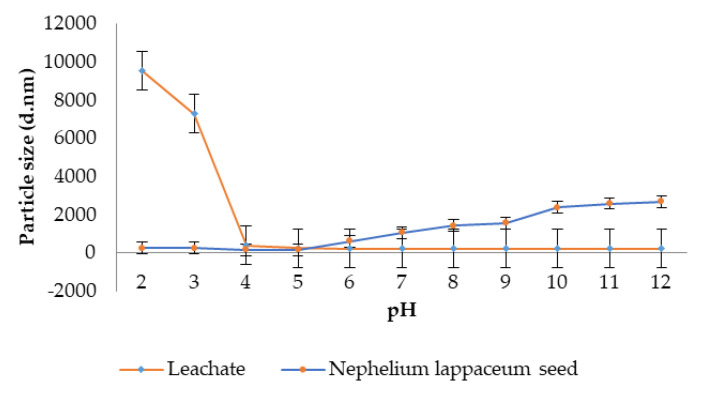
pH effect on particle size of leachate.

**Figure 7 ijerph-19-00420-f007:**
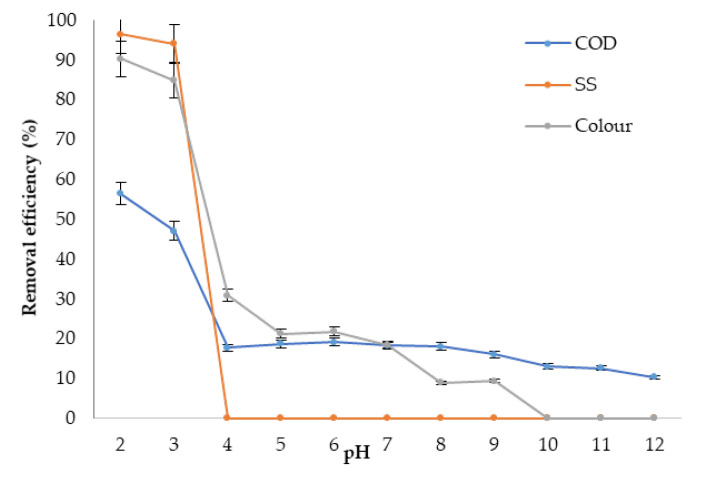
Effect of pH on pollutant removal using 1.5 g/L of *Nephelium lappaceum* seed dosage.

**Figure 8 ijerph-19-00420-f008:**
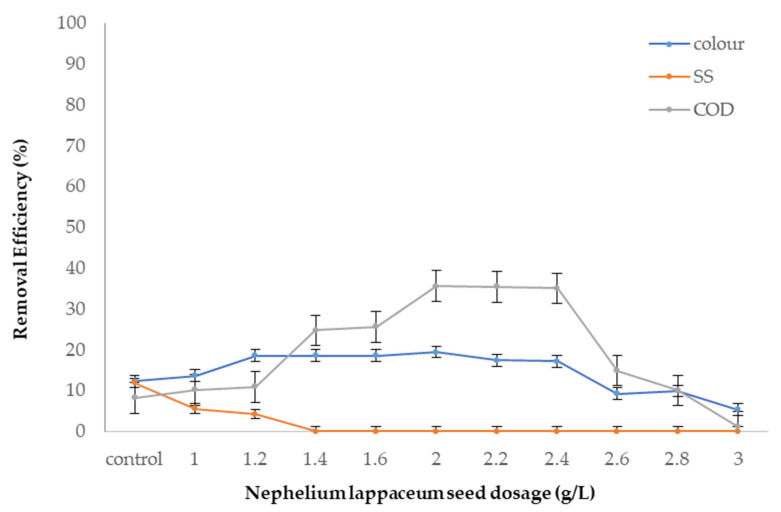
SS, colour, and COD removal efficiency with different dosages of *Nephelium lappaceum* seed at pH 6.

**Figure 9 ijerph-19-00420-f009:**
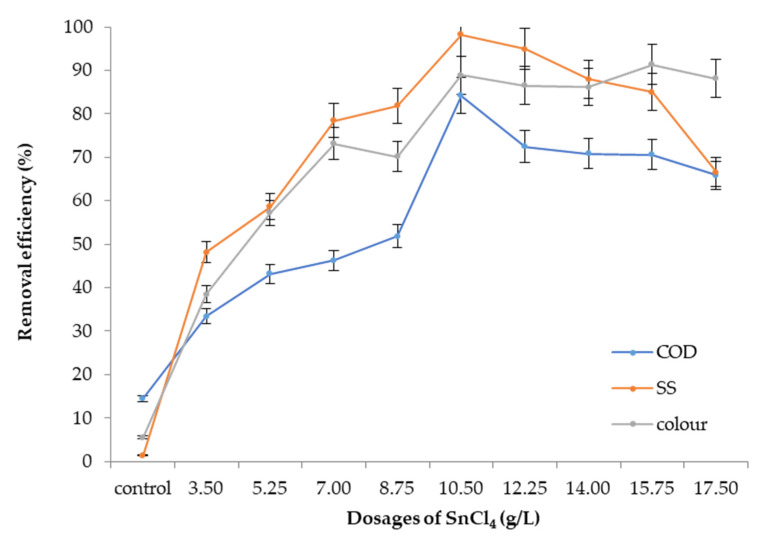
SS, colour, and COD removal efficiency with varied SnCl_4_ dosages at optimum pH 7.

**Figure 10 ijerph-19-00420-f010:**
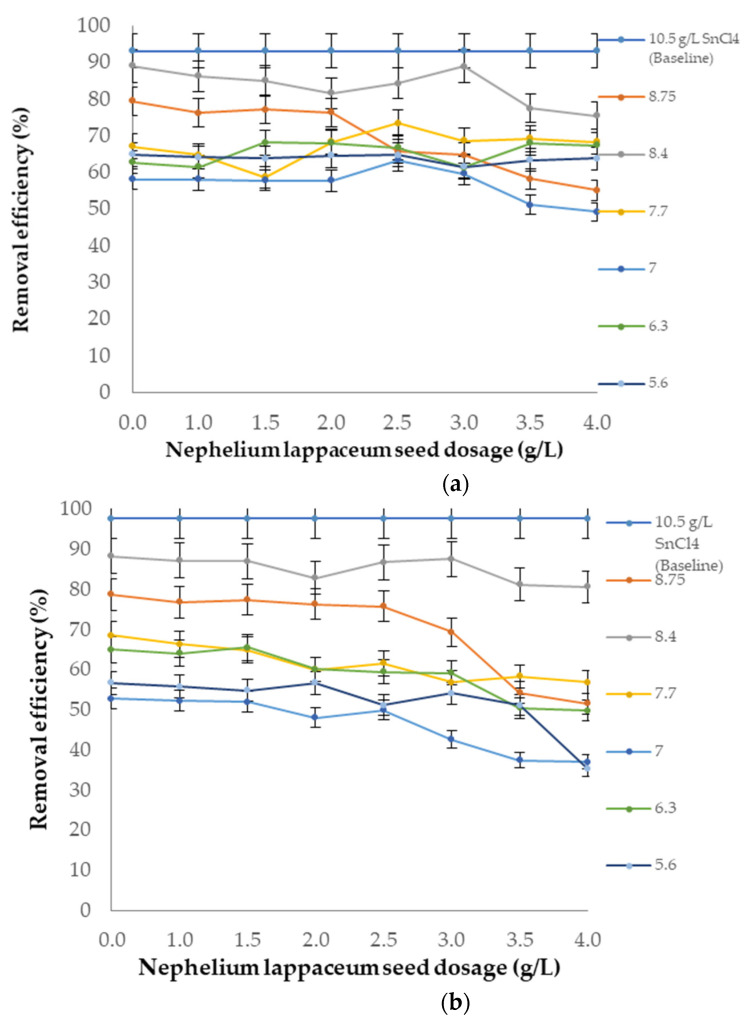
Removal efficiencies of pollutants at pH 7 with varied dosages of SnCl_4_ as coagulant and *Nephelium lappaceum* seed as a flocculant. (**a**) Colour; (**b**) Suspended solids (SS); (**c**) COD.

**Figure 11 ijerph-19-00420-f011:**
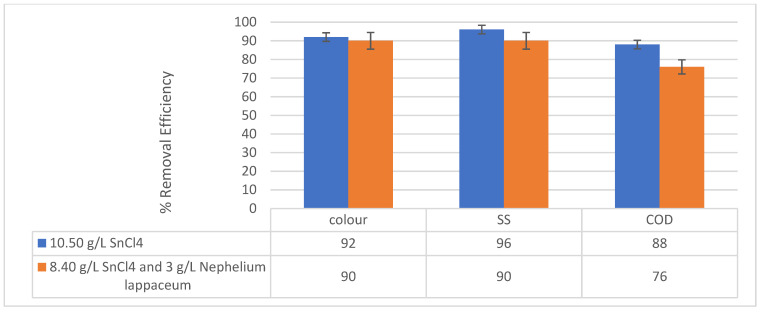
A summary of the removal efficiencies of pollutants conducted at pH 7 with varied dosages of SnCl_4_ as coagulant and *Nephelium lappaceum* seed as a flocculant.

**Table 1 ijerph-19-00420-t001:** Alor Pongsu Landfill Site (APLS) raw leachate characteristics.

Parameter	Min	Max	Average *	Permissible Limit
Temperature (°C)	26.48	32.07	29.47	40
pH	8.04	8.90	8.59	6.0–9.0
Total Dissolved Solids (g/L)	8.532	9.950	9.440	
Dissolved Oxygen (mg/L)	2.14	4.38	3.17	
BOD5 (mg/L)	160	333	241	20
COD (mg/L)	2533	4215	3440	400
Ratio BOD_5_/COD	0.057	0.080	0.070	
Suspended Solids (mg/L)	350	604	469	50
Colour (PtCo)	13750	18733	16829	100
NH_3_-N	1040	1357	1227	5
Zeta Potential (mV)	−22.4	−20.7	−21.5	

* Average value for six samples taken from January to April 2020.

**Table 2 ijerph-19-00420-t002:** Functional groups of *Nephelium lappaceum* seed.

Wavelength (cm^−1^)	Wavenumber Range	Intensity	Bond	Group Vibration	Functional Group
3585	3700–3584	Medium	O–H	Stretching	Alcohol
3564	3700–3500	Medium	N–H	Stretching	Amide
3544	3550–3200	Strong	O–H	Stretching	Alcohol
3367	3400–3300	Medium	N–H	Stretching	Aliphatic primary amine
3307	3300–2500	Strong	O–H	Stretching	Carboxylic acid
3004	3000–2800	Strong	N–H	Stretching	Amine salt
	3100–3000	Medium	C–H	Stretching	Alkene
2917	3000–2800	Strong	N–H	Stretching	Amine salt
2850	3000–2840	Medium	C–H	Stretching	Alkane
2307	2280–2440	Medium	P–H	Stretching	Phosphine
1735	1750–1735	Strong	C=O	Stretching	δ–lactone
1712	1725–1705	Strong	C=O	Stretching	Aliphatic ketone
	1720–1706	Strong	C=O	Stretching	Carboxylic acid
	1650–1580	Medium	N–H	Bending	Amine
	1650–1566	Medium	C=C	Stretching	Cyclic alkene
1506	1550–1500	Strong	N–O	Stretching	Nitro compound
1471	1400–1480	Strong	C–O	Stretching	Methylene
1416	1440–1395	Medium	O–H	Bending	Carboxylic acid
	1420–1330	Medium	O–H	Bending	Alcohol
1384	1415–1380	Strong	S=O	Stretching	Sulfate
	1410–1380	Strong	S=O	Stretching	Sulfonyl chloride
	1400–1000	Strong	C–F	Stretching	Fluoro compound
1249	1342–1266	Strong	C–N	Stretching	Aromatic amine
	1250–1020	Medium	C–N	Stretching	Amine
1209	1275–1200	Strong	C–O	Stretching	Alkyl aryl ether
	1225–1200	Strong	C–O	Stretching	Vinyl ether
	1210–1163	Strong	C–O	Stretching	Ester
1155	1170–1155	Strong	S=O	Stretching	Sulphonamide
	1165–1150	Strong	S=O	Stretching	Sulfonic acid
	1160–1120	Strong	S=O	Stretching	Sulfone
	1205–1124	Strong	C–O	Stretching	Tertiary alcohol
1078	1085–1050	Strong	C–O	Stretching	Primary alcohol
1022	1050–1040	Strong	CO–O–CO	Stretching	Anhydride
861	880 ± 20	Strong	C–H	Bending	1,2,4—trisubstituted
	880 ± 20	Strong	C–H	Bending	1,3—disubstituted
762	850–550	Strong	C–Cl	Stretching	Halo compound
	780 ± 20	Strong	C–H	Bending	1,2,3—trisubstituted
	755 ± 20	Strong	C–H	Bending	1,2—disubstituted
	750 ± 20	Strong	C–H	Bending	Monosubstituted benzene derivative
718	730–665	Strong	C=C	Bending	Alkene
601	850–550	Strong	C–C	Stretching	Halo compound
	690–515	Strong	C–Br	Stretching	Halo compound
574	850–550	Strong	C–Cl	Stretching	Halo compound
534	850–550	Strong	C–Cl	Stretching	Halo compound
	600–500	Strong	C–I	Stretching	Halo compound

**Table 3 ijerph-19-00420-t003:** Comparison of previous and current studies for best pH values of various natural coagulant.

Natural Coagulant	Optimum pH	Pollutant Removal Rate (%)			Reference
		SS	Colour	COD	
Commercial sago starch (CSS)	4	29.5	15.1	28.0	[37]
Chitosan	4	-	14.7	-	[32]
Durian seed starch	6	-	34.0	-	[45]
Tamarindus indica seed (TiS)	4	-	30.2	7.5	[46]
Longan seed	4	-	10.3	5.2	[19]
**Nephelium lappaceum seed**	**6**	**-**	**21.8**	**19.2**	**Current Study**

The bold is to show the results of the current study.

**Table 4 ijerph-19-00420-t004:** Comparison of optimum dosage of *Nephelium lappaceum* with other natural coagulants from previous studies.

Natural Coagulant.	Optimum Dosage (g/L)	Pollutant Removal Rate (%)			Source
		SS	Colour	COD	
Commercial sago starch	6	29.5	15.1	28	[37]
Chitosan	0.06	-	14.7	-	[32]
Durian seed starch	4	-	34	-	[45]
Tamarindus indica seed	5.00	-	41.90	5.90	[46]
Longan seed	2.00	29.5	15.10	28.00	[19]
**Nephelium lappaceum seed**	**2.00**	**-**	**19.48**	**35.62**	**Current Study**

The bold is to show the results of the current study.

## Data Availability

The data presented in this study are available on request from the corresponding author.

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
