# Peer review of "The Potential Use of Nephelium lappaceum Seed as Coagulant–Coagulant Aid in the Treatment of Semi-Aerobic Landfill Leachate"

_ijerph, 2021, doi:10.3390/ijerph19010420_

Round 1
Reviewer 1 Report
This is an interesting attempt of utilizing Nephelium lappaceum seed as coagulant aid in the treatment of semi-aerobic landfill leachate.
Manuscript is very well-written and minor revision is recommended for acceptance.
The most important issue for me is that I would like in the Introduction if anything is mentioned about the coagulant SnCl4, as well as what is the reason we are looking for coagulant aid. Also, the results show smaller removals when used SnCl4 + Nephelium.
Here are my specific comments:
- Page 3, line 104: Please replace the word was with the word were.
- Page 3, lines 131-132: Please explain which is the necessary pH?
- Page 5, line 190: Correct the word Low with low.
- Page 7, Table 2 is not displayed correctly.
- Page 14, line 483: Delete Th at the beginning of the sentence.
Author Response
Please refer to the attached response file
Reviewer 2 Report
The potential use of Nephelium lappaceum seed as coagulant-coagulant aid in the treatment of semi-aerobic landfill leachate
This study examines the potential application of Nephelium lappaceum seed, a natural co-13 agulant- coagulant aid with Tin (IV) chloride (SnCl4) in eliminating suspended solids (SS), colour, 14 and chemical oxygen demand (COD) from landfill leachate. The manuscript needs a revision to be reconsidered for publication. Therefore, in my point of view, the manuscript could be considered for acceptance but not in its current form. Having said that following revisions are suggested;
Comments:
The abstract is vague.
Normally an abstract should state briefly the purpose of the study undertaken and meaningful conclusions based on the obtained results. Hence, this needs rewriting. I would expect brief, yet concise, the quantitative data description of the results in the abstract.
The novelty of the study should be clearly highlighted in the manuscript at the end of the introduction section, as there are some existing literature reports.
Did the authors conduct the repeated experiments in your study? If you did, please show the repeated experiments and the analysis of statistical significance of differences in Material and methods section!
Moreover, you should present the analysis of statistical significance of differences in the results section.
The manuscript should be carefully revised so that the results are better discussed. In my opinion, authors mainly focused on results and the Discussion section lacks scientific depth.
All sections should be critically discussed and compared with the previous reports. This will actually strengthen the manuscript and will highlight the significance of the work.
The conclusion is superficial. Herein, I would like to see the major findings and how they are addressing the left behind research gaps and covering current challenges.
The level of English used is not up to the journal standard. Throughout the manuscript, the level of English used is not up to the standard of the journal.
Explain APLS in the table 1 caption.
Statistical analysis is missing in table 1.
Table 2 expression is not correct.
Figure 5, all error bars look the same, please check it
Figure 6, Why there is a high error bars in the case of leachate at pH 4-12, even value is zero. There is something wrong.
Figure 8, again error bars are same. Authors did not perform this carefully. I depreciate this approach.
Fig 9, and 10, same problem.
Author Response

(The authors gave the same response as above.)

Reviewer 3 Report
The authors of this paper presented an very interesting article on the use of Nephelium lappaceum seed as an alternative coagulant in leachate treatment in combination with tin (IV) chloride which acted as the main coagulant. I suggest editorial corrections to the graphical parts of the paper (graphs) for easier reading. The paper is summarised with appropriate conclusions presenting the results of the research carried out.
I provide a list of minor comments below:
Line 42, space delete before "On the"
Line 51 space delete before "Furthermore"
Line 60 delete spaces before "The mechanism".
Line 82 space delete before "specifically".
Table 1 BOD5 replace with BOD5
Figure 1, what does (a) in brackets mean?
Table 2 is unreadable
Figure 4 either a) or (a)
Equation 1 should be more clearly described as to what Ci is and what Cf is, and separated from the text.
Line 248 Remove space before "Figure"
Add axis marks in Figures 5,6,7,8,10 for easier reading
Line 339 remove space before "Beyond"
Figure 7, line markings 346-358 enter the axis of the figure.
Figure 9, line markings 406-415 enter the axis of the figure.
Author Response

(The authors gave the same response as above.)

Round 2
Reviewer 2 Report
Required changes made by the author, hence I am recommending this manuscript for publication.